# Comparison of the change in QuantiFERON-TB Gold Plus and QuantiFERON-TB Gold In-Tube results after preventive therapy for latent tuberculosis infection

Ock-Hwa Kim[1], Kyung-Wook Jo[1], Shinhee Park[2], Yong-Ha Jo[3], Mi-Na Kim[4], Heungsup Sung[4], Tae Sun Shim[1] *

1 Division of Pulmonology and Critical Care Medicine, University of Ulsan College of Medicine, Asan Medical Center, Seoul, South Korea, 2 Department of Pulmonary, Allergy, and Critical Care Medicine, University of Ulsan College of Medicine, Gangneung Asan Hospital, Gangneung, South Korea, 3 University of Ulsan College of Medicine, Seoul, South Korea, 4 Department of Laboratory Medicine, University of Ulsan College of Medicine, Asan Medical Center, Seoul, South Korea

☯ These authors contributed equally to this work.
* shimts@amc.seoul.kr

**Data Availability Statement:** All relevant data are within the manuscript and its Supporting Information files.

## Abstract

### Background

We investigated changes in the interferon-γ levels before and after treatment of latent tuberculosis infection (LTBI) using QuantiFERON-TB Gold Plus (QFT-Plus) and QuantiFERON-TB Gold In-Tube (QFT-GIT) assays. The objective was to assess whether QFT-Plus could serve as a biomarker of LTBI treatment response.

### Methods

We prospectively enrolled 44 individuals whose baseline QFT-GIT and QFT-Plus showed positive results at a tertiary referral center in South Korea between March 2017 and March 2018. The results of the QFT-Plus assay were defined as positive if either or both of the antigen tubes (TB1 and/or TB2) were positive. After LTBI treatment, both tests were repeated.

### Results

The mean age of the participants was 47.6 years. The QFT-GIT and QFT-Plus assays revealed positive results in 42/44 (95.5%) and 41/44 (93.2%) participants after LTBI treatment, showing overall agreement of 93.2%, with a Cohen's kappa value of 0.37 (fair agreement). The differences between pre- and post-LTBI treatment interferon-γ levels were measured using the QFT-GIT and QFT-Plus assays. No significant differences were noted among the 3 values: the median difference in interferon-γ value with QFT-GIT, QFT-Plus TB1, and QFT-Plus TB2 was 0.211 IU/mL (IQR, −0.337–3.347), 0.025 IU/mL (IQR, −0.338–1.368), and 0.180 IU/mL (IQR, −0.490–2.278), respectively (P = 0.401).

**Funding:** The authors received no specific funding for this work.

**Competing interests:** The authors have declared that no competing interests exist.

## Conclusion

The change in interferon-γ levels before and after LTBI treatment measured using the QFT-Plus assay showed a similar trend to that of the QFT-GIT assay. Considering that the QFT-GIT assay is not a useful biomarker of LTBI treatment response, QFT-Plus also appears not to be useful for this purpose.

## Introduction

Latent tuberculosis infection (LTBI) is defined as a state wherein humans are infected with *Mycobacterium tuberculosis* without any clinical symptoms, radiological abnormality, or microbiological evidence of infection [1,2]. According to the World Health Organization, approximately 2–3 billion people in the world are affected by LTBI [3]. Given that approximately 5%–10% of immunocompetent persons with LTBI will develop overt tuberculosis (TB) in their lifetime [4], treatment of LTBI has direct implications for global prevention and control efforts against TB.

The World Health Organization guidelines recommend the use of a tuberculin skin test or interferon-γ release assay (IGRA) for detection of LTBI in high-income and upper middle-income countries that have an estimated TB incidence of <100 per 100,000 population [5]. Until recently, two IGRAs have been commercialized: the T-SPOT.TB assay (Oxford Immunotec, Abingdon, United Kingdom) and the QuantiFERON-TB Gold In-Tube (QFT-GIT) assay (Cellestis/Qiagen, Carnegie, Australia) [6,7]. The QFT-GIT assay measures interferon-γ (IFN-γ) concentration to detect *M. tuberculosis* infection using a single mixture of synthetic peptides (ESAT-6, CFP-10, and TB7.7) to stimulate T cells [6]. Although the ESAT-6 and CFP-10 antigens stimulate both CD4$^+$ and CD8$^+$ T cells to release IFN-γ, these two epitopes mainly stimulate CD4$^+$ T cells [8]. However, when exposed to *M. tuberculosis* antigens, both CD4$^+$ and CD8$^+$ T cells secrete IFN-γ [9]. Therefore, the newest generation of the QFT assay, QuantiFERON-TB Gold Plus (QFT-Plus) was recently developed and launched in 2015. In contrast to QFT-GIT assay, QFT-Plus has two TB antigen tubes: TB antigen tube 1 (TB1) and TB antigen tube 2 (TB2). TB1 contains long peptides from ESAT-6 and CFP-10, which are designed to stimulate an immune response from CD4$^+$ T cells. In contrast, TB2 contains relatively short peptides that elicit immune responses from both CD4$^+$ and CD8$^+$ T cells. The TB7.7 present in the QFT-GIT has been removed in the QFT-Plus assay [8,10].

One of the most frustrating problems in the treatment of patients with LTBI is that the IGRA results are not a surrogate marker of response to LTBI therapy. Previous studies have shown no differences in the decline in QFT-GIT or reversion to negative results between patients who received LTBI treatment and those who received placebo or no treatment [11,12]. Given the reported positive correlation between the CD8$^+$ T-cell response against TB antigens and higher CD8$^+$ T-cell responses in the context of recent exposure to *M. tuberculosis* [13,14], we postulated that the CD8$^+$ response would be more sensitive to LTBI treatment compared with the CD4$^+$ response. Therefore, in this study, we examined the changes in IFN-γ levels as measured using QFT-Plus and QFT-GIT assays before and after LTBI treatment. The objective was to assess whether QFT-Plus can be employed as a useful biomarker of LTBI treatment response.

## Materials and methods

### Study population

This prospective trial was conducted between March 2017 and March 2018 at a tertiary referral center in South Korea, where the incidence of TB is intermediate (80–100/100,000 people per

year) [15]. Adults aged 18–80 years, who were candidates for an LTBI screening test for various reasons (such as healthcare personnel undergoing screening, contact investigation, and candidate recipients of tumor necrosis factor inhibitor therapy) were prospectively recruited. All enrolled participants were examined using the following strategy to rule out the possibility of active TB. All participants were interviewed by the attending physician regarding the presence of any respiratory symptoms that were suggestive of active TB, such as cough, sputum, or hemoptysis, in addition to undergoing a simple chest radiography (CXR). For those with radiographic anomalies suggestive of TB, further evaluation was performed using sputum acid-fast bacilli smear and culture to rule out active TB or lung disease due to nontuberculous mycobacteria. Moreover, present and former CXR findings were compared to identify the presence of newly developed lesions.

At baseline, both QFT-GIT and QFT-Plus were performed for all participants. Then, among the participants with a positive QFT-GIT assay, those whose QFT-Plus assay (TB antigen tube 1 [TB1] or TB antigen tube 2 [TB2]) results were also positive were selected for the present study. After the completion of LTBI treatment, both tests were performed again for all participants to assess the post-treatment changes in IFN-γ level. The exclusion criteria were as follows: (i) those whose baseline QFT-GIT and/or QFT-Plus tests (both TB1 and TB2) revealed a negative result; (ii) those who did not receive QFT tests after LTBI treatment; (iii) participants who did not complete the treatment; and (iv) participants with an indeterminate QFT result. The trial protocol was approved by the institutional review board (IRB) at the Asan Medical Center (IRB No.: 2017–0004). Written informed consent was obtained from all participants before enrollment.

## The QFT test assay

The QFT-Plus kits were donated by Qiagen and were used according to the manufacturer's instructions at our center [16]. QFT-GIT was also performed at our center as per the manufacturer's instructions [17]. We defined QFT-GIT results as positive (1) if the nil value was ≤8.0 IU/mL, (2) the TB antigen minus the nil IFN-γ value was ≥0.35 IU/mL, and (3) at least 25% of the nil value, irrespective of the mitogen minus nil value [18]. The same criteria applied to the QFT-Plus assay. However, the results of the QFT-Plus assay were defined as positive if either or both of the TB antigen tubes (TB1 and/or TB2) were positive [18]. If antigen-nil was <0.35 IU/mL or <25% of the nil value, when the mitogen was ≥0.5 IU/mL, the result was considered negative. If (1) nil was >8 IU/mL or (2) antigen-nil ≥0.35 IU/mL and <25% of the nil value when the nil was ≤8.0 IU/mL and the mitogen was <0.5 IU/mL, the results were considered indeterminate. The IFN-γ levels in the antigen tubes >10 IU/mL were classified as 10 IU/mL.

## LTBI treatment regimen

During the study period, daily isoniazid and rifampin for 3 months were selected as the primary regimen for LTBI treatment. Based on the discretion of the attending physician, a regimen of rifampin for 4 months could also be prescribed. Treatment completion was defined as intake of >80% of all prescribed medication within 4 months for isoniazid and rifampin and within 6 months for rifampin.

## Statistical analysis

All analyses were performed using SPSS software (version 20.0; SPSS, Chicago, IL, USA). The data are presented as mean (± standard deviation) or median (interquartile range [IQR]) for continuous variables and as percent for discrete variables. The Friedman test was used to compare the IFN-γ levels measured in different antigen-containing tubes before LTBI treatment

and changes in the paired IFN-γ levels before and after LTBI treatment. In cases where there was a significant difference among the three values, a *post hoc* analysis using the Wilcoxon signed rank test for multiple comparisons was performed with the application of Bonferroni correction.

To qualitatively compare the results of the QFT-GIT and QFT-Plus assays after LTBI treatment, positive, negative, and overall percent agreements were calculated using the QFT-GIT results as the reference standard. Concordance between the QFT-GIT and QFT-Plus assays was assessed using kappa coefficients; kappa values of ≤0, 0.01–0.20, 0.21–0.40, 0.41–0.60, 0.61–0.80, and ≥0.81 were interpreted to indicate no, slight, fair, moderate, substantial, and almost perfect agreement, respectively [19].

## Results

### Study participants

We enrolled 44 participants whose baseline QFT-GIT and QFT-Plus assay showed positive results and who received repeat tests of the QFT-GIT and QFT-Plus assay after LTBI treatment (Fig 1). The mean age was 47.6 years. The most common indication for LTBI testing was baseline or periodic screening tests for healthcare workers, followed by LTBI screening for the day care center workers. Almost all (42/44, 95.5%) of the participants completed treatment with isoniazid and rifampin for 3 months, without any change in the regimen. The baseline characteristics of the 44 participants are shown in Table 1. Among the 44 participants, most (41/44, 93.2%) were positive with both QFT-Plus TB1 and TB2, with the exception of 3 participants, 1 of who was positive with QFT-Plus TB1 only, whereas the other 2 were positive with QFT-Plus TB2 only (Table 2).

---

**98** LTBI participants with QFT-GIT positive results between March 2017 and March 2018 (n = 98)

Excluded (n = 22)
  QFT-Plus TB1 and TB2 negative results (n = 22)

**76** participants with both QFT-GIT and QFT-Plus positive results before treatment (n = 76)

Excluded (n = 31)
  Did not perform QFT test after treatment (n = 27)
  Treatment discontinued (n = 2)
  Not treated (n = 2)

**45** LTBI participants with available QFT-GIT and QFT-Plus data before and after treatment (n = 45)

Excluded (n =1)
  Indeterminate results with QFT-GIT and
  QFT-Plus after treatment (n = 1)

**44** LTBI participants with comparable QFT-GIT and QFT-Plus data before and after treatment (n = 44)

**Fig 1. Study flow chart.** LTBI, latent tuberculosis infection; QFT-GIT, QuantiFERON-TB Gold In-Tube; QFT-Plus, QuantiFERON-TB Gold Plus.

---

**Table 1. Clinical characteristics of the study participants.**

| Characteristics | Total (n = 44) |
|---|---|
| Age (years) | 47.6 ± 11.4 |
| Female sex | 29 (65.9%) |
| Body mass index at initiation of treatment (kg/m$^2$) | 23.6 ± 3.1 |
| Current or past smoker | 9 (20.5%) |
| Previous history of TB | 0 (0.0%) |
| Indication for LTBI treatment | |
| Health care workers | 19 (43.2%) |
| Day care center workers | 10 (22.7%) |
| Treatment with TNF-α inhibitor | 7 (15.9%) |
| Close contact with persons with TB | 4 (9.1%) |
| CXR findings suggestive of healed TB | 3 (6.8%) |
| Recent TB infection (<2 years) | 1 (2.3%) |
| Treatment regimen | |
| 3 months of isoniazid and rifampin | 42 (95.5%) |
| 4 months of rifampin | 2 (4.5%) |

Abbreviations: TB, tuberculosis; LTBI, latent tuberculosis infection; TNF, tumor necrosis factor; CXR, chest X-ray.

Data are reported as mean (± standard deviation) or as frequencies (%).

## Baseline QFT-GIT and QFT-Plus value

Fig 2 shows the IFN-γ levels of 44 participants with LTBI measured using the QFT-GIT and QFT-Plus (TB1 or TB2) assay before preventive treatment. Overall, statistical significance was noted among the three baseline IFN-γ values. That is, the median IFN-γ level value of the QFT-GIT assay was slightly higher than that of the QFT-Plus assay; the median IFN-γ level of QFT-GIT was 3.395 IU/mL (IQR, 1.615–7.180), the median TB1 QFT-Plus antigen IFN-γ level was 3.060 IU/mL (IQR, 0.655–7.045), and the median TB2 QFT-Plus antigen IFN-γ level was 2.880 IU/mL (IQR, 0.668–7.170) ($P = 0.040$). However, the difference was not statistically significant between the QFT-GIT and QFT-Plus TB1 ($Z = -1.743$, $P = 0.081$) or between the QFT-GIT and QFT-Plus TB2 ($Z = -1.367$, $P = 0.172$) assays.

## The results of QFT assays after completion of LTBI treatment

After LTBI treatment, the QFT-GIT and QFT-Plus assays revealed positive results in 42/44 (95.5%) and 41/44 (93.2%) participants. Among the 44 participants with positive QFT-Plus assay results at baseline, 41/44 (93.2%) responded to both TB1 and TB2 at baseline, and the proportion did not change after treatment completion. In the remaining 3 participants, the QFT-Plus assay reverted to negative after LTBI treatment. The detailed results of the QFT-Plus according to the participants who responded to TB1 and/or TB2 peptides are shown in Table 2.

## Comparison of the QFT-GIT and QFT-Plus assays after completion of LTBI treatment

Then, we qualitatively compared the results of the QFT-GIT and QFT-Plus assays after completion of LTBI treatment. The QFT-Plus assay showed agreement with the QFT-GIT assay in 40/42 (95.2%) positive tests and 1/2 (50.0%) negative tests and had an overall agreement of 41/44 (93.2%) among all participants, with a Cohen's kappa value of 0.37 (fair agreement).

**Table 2. The results of QFT assays at baseline and after the completion of preventive therapy.**

| | At baseline | After LTBI treatment |
| --- | --- | --- |
| | Positive N (%) | Positive N (%) |
| QFT-GIT | 44 (100) | 42 (95.5) |
| QFT-Plus TB1 or TB2 | 44 (100) | 41 (93.2) |
| QFT-Plus TB1 and TB2 | 41 (93.2) | 41 (93.2) |
| QFT-Plus only TB1 | 1 (2.3) | 0 (0) |
| QFT-Plus only TB2 | 2 (4.5) | 0 (0) |

Abbreviations: QFT, QuantiFERON; LTBI, latent tuberculosis infection; QFT-GIT, QuantiFERON-TB Gold In-Tube; QFT-Plus, QuantiFERON-TB Gold Plus.

Among the 44 participants enrolled in our study, the QFT-GIT and QFT-Plus assays after completion of LTBI treatment were discordant in 3 (6.8%) participants (Table 3).

## Changes in IFN-γ levels of two QFT assays after completion of LTBI treatment

Next, we calculated the differences between the pre- and post-LTBI treatment IFN-γ levels (post-treatment IFN-γ level minus pre-treatment IFN-γ level), measured using the QFT-GIT and QFT-Plus assays. The median duration from baseline QFT tests to initiation of LTBI treatment was 42 days (IQR, 26–65). The follow-up QFT tests were performed at a median interval of 2 days (IQR, 0–8.8) after the completion of LTBI treatment. As shown in Fig 3, no significant differences were noted among the three values. The median difference in the IFN-γ value of QFT-GIT was 0.211 IU/mL (IQR, −0.337–3.347), the median difference in TB1 QFT-Plus antigen IFN-γ level was 0.025 IU/mL (IQR, −0.338–1.368), and the median difference in TB2 QFT-Plus antigen IFN-γ level was 0.180 IU/mL (IQR, −0.490–2.278) (*P* = 0.401). We also evaluated the overall median IFN-γ levels before and after treatment, the results of which are shown in the supporting information (S1 Fig and S1 Table). In addition, when we compared the IFN-γ response to TB1 and TB2 peptides at the same time point, we observed similar levels

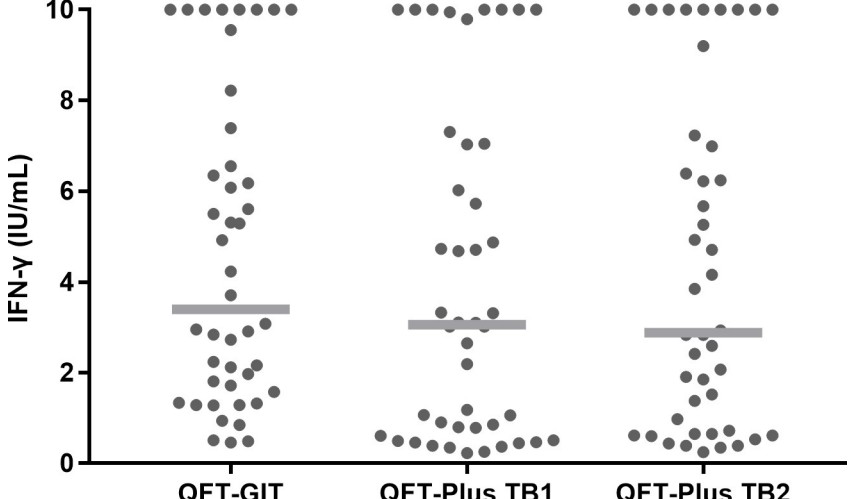

**Fig 2. IFN-γ levels measured using different antigen-containing tubes before preventive therapy.** IFN-γ concentrations from each participant are shown (•). Median levels are indicated by the gray lines. IFN-γ, interferon-γ; QFT-GIT, QuantiFERON-TB Gold In-Tube; QFT-Plus, QuantiFERON-TB Gold Plus.

**Table 3. Qualitative comparison of QFT-GIT and QFT-Plus assays after preventive therapy.**

| QFT-Plus result | QFT-GIT result | | % agreement (95% CI) | | | Kappa value |
|---|---|---|---|---|---|---|
| | Positive | Negative | Positive | Negative | Overall | |
| Positive | 40 | 1 | 95.2 (83.4–99.5) | 50.0 (9.5–90.6) | 93.2 (81.1–98.3) | 0.37 (−0.19–0.92) |
| Negative | 2 | 1 | | | | |

Abbreviations: QFT-GIT, QuantiFERON-TB Gold In-Tube; QFT-Plus, QuantiFERON-TB Gold Plus.

of IFN-γ and a positive correlation between IFN-γ values in response to TB1 and TB2 stimulation both at baseline and after LTBI treatment (S2 Fig).

## Discussion

Since the QFT-Plus has been developed, several studies have compared the performance of QFT-GIT and QFT-Plus for the diagnosis of active TB [20], contact screening [10], LTBI [21], and screening of healthcare personnel [18]. However, no study has yet analyzed the QFT-Plus responses before and after LTBI treatment compared with the QFT-GIT assay result. To the best of our knowledge, this is the first study that evaluates this issue. The most important finding of the present study was that the trend of changes in the IFN-γ levels before and after LTBI treatment measured using the QFT-Plus assay (either TB1 or TB2) was similar to that observed

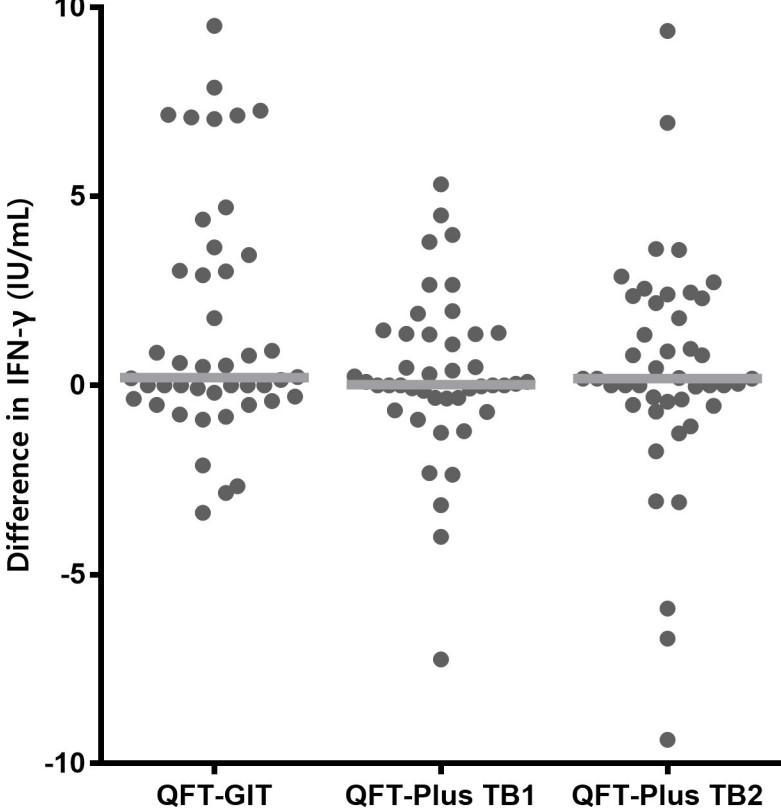

**Fig 3. Differences in IFN-γ levels before and after preventive therapy.** The differences in IFN-γ concentrations before and after preventive therapy for each participant are shown (•). Median levels are indicated by gray lines. IFN-γ, interferon-γ; QFT-GIT, QuantiFERON-TB Gold In-Tube; QFT-Plus, QuantiFERON-TB Gold Plus.

with the QFT-GIT assay. Our findings suggest that QFT-Plus is unlikely to be a useful biomarker of LTBI treatment response, based on earlier reports [11,22] that showed that the QFT-GIT cannot be used as the biomarker of LTBI treatment response.

Previous studies have reported that IGRA is probably not a useful biomarker of treatment efficacy in the context of LTBI, because LTBI treatment appears to have no effect on the results of serial IGRA testing. In their prospective trial involving 82 patients with LTBI, Johnson et al. had found no differences with respect to the decline in QFT-GIT or reversion to negative results after isoniazid preventive therapy or with observation alone [11]. Another placebo-controlled randomized trial revealed no differences in IFN-γ enzyme-linked immunosorbent spot test (ELISPOT) reversion over time between the treatment and placebo arms [12]. Bastos et al. also reported frequent within-participant variability in the QFT-GIT responses, which was not associated with LTBI treatment in high-risk tuberculin skin test-positive participants [22]. The present study compared the difference in the IFN-γ value measured using the QFT-Plus assay with the QFT-GIT result, without using a placebo (or observation) arm. However, given QFT-GIT is known to have no usefulness as a marker for LTBI treatment [11,22], and the differences between pre- and post-treatment IFN-γ values showed a similar trend between QFT-GIT and QFT-Plus (TB1 or TB2) assays in the present study, we can conclude that QFT-Plus does not appear to be a useful biomarker either.

In the case of active TB, previous studies have shown that the *M. tuberculosis* specific CD8+ T-cell response declines with anti-TB treatment and could be a surrogate marker of anti-TB therapy response [23,24]. In a prospective cohort study of 50 patients with culture-confirmed TB, a significant difference was observed in the *M. tuberculosis* specific CD8+ T-cell response after anti-TB treatment, but not in the CD4+ T-cell response [23]. This decline in response was noted over 24 weeks of treatment, with the earliest change observed at 8 weeks of therapy. In addition, Kamada et al. had suggested that QFT-Plus could be used as a potential tool for monitoring the efficacy of anti-TB therapy by showing a decline in the surrogate CD8+ T-cell response (difference "TB2 minus TB1") at the end of treatment [24]. A recently published study reported a reduction in IFN-γ production over time in response to both TB1- and TB2-peptide stimulation in 28 patients with active TB [25]. Conversely, in the present study, we did not note any significant differences between the baseline and post-treatment TB1 and TB2 response in participants with LTBI (Fig 3). We believe that the difference in response between active TB and LTBI can be attributed to the differences with respect to the mycobacterial load [26] or the persistence of immunoreactive cells due to the persistence of antigen or mycobacteria after LTBI treatment [27].

Our findings are in contrast with those of a recent study by Petruccioli et al. in which LTBI therapy significantly reduced IFN-γ levels and the number of responders to TB1- and TB2-peptide stimulation [25]. The most plausible explanation for this disparity could be differences in the patient population. In the study by Petruccioli et al., 80.3% (37/46) of patients with LTBI had been recent contacts of patients with pulmonary TB, whereas the majority of participants in our study were healthcare and day care center workers or candidates for TNF inhibitor therapy. That is, only 11.4% (5/44, 4 in close contact with people with TB and 1 recently converted, Table 1) were likely to have been infected recently. Given that another previous study had also shown that the likelihood of tuberculin skin test reversion to negative following treatment with isoniazid in a group of US Navy personnel was strongly associated with the recency of infection [28], different results might have been observed if we mainly enrolled participants who were likely to have been infected recently. Although the findings were generally the same when we stratified our participants according to the possibility of recent infection (S3 Fig), it was likely that the number of recently infected participants was too small to reveal statistical significance.

Although we generally did not observe a statistically significant trend in terms of changes in IFN-γ level before and after LTBI treatment, some participants had a reduction in the TB1 and/or TB2 response on follow-up tests. This decline could be attributed to the natural course of IFN-γ response rather than to the effects of LTBI treatment, considering the findings of a previous study showing that approximately half of the contacts of patients with TB with initial positive ELISPOT assay had shown a negative result on repeat ELISPOT assay at 3 months, in the absence of treatment [29].

IFN-γ response failed to distinguish between active infection and cleared infection because of the persistence of antigen-specific memory T cells induced by prior exposure to *M. tuberculosis* [30]. In the present study, as depicted in Fig 2, we frequently observed increases in IFN-γ levels. This increase could have been due to continued exposure to *M. tuberculosis* in a high-incidence setting or persistent infection with latent bacilli. We also noted a post-treatment decline in IFN-γ levels in some of our study participants, which could have been caused by a natural decline in T-cell frequencies [12,31]. These factors collectively explain the reason why IGRA is of no use as a biomarker of infection clearance efficacy following LTBI treatment in previous studies as well as the present one. The well-documented inherent variability of the IGRA test is another important factor to be considered [32]. Considering this inherent IGRA test variability, we performed our analysis again after stratifying our participants according to the baseline IFN-γ value of QFT-Plus (either TB1 or TB2) as previously described [25]: 0.2–0.34 IU/mL; 0.35–0.7 IU/mL; and >0.7 IU/mL. The overall results are shown in the supporting information (S4 Fig). Reversions tended to be more in the uncertainty zone, which had been defined as baseline IFN-γ value 0.2–0.7 IU/mL in previous studies [25,33]. This result suggests that some of the reversions could be attributable to inherent variability.

As shown in Fig 2, the stimulated IFN-γ values obtained with QFT-Plus were statistically significantly lower than those obtained with the QFT-GIT assay. This finding was consistent with those of previous studies [8,20] and is probably because the QFT-Plus assay released less IFN-γ than the QFT-GIT assay because the QFT-Plus assay does not contain TB7.7. A previous study had shown that the epitope TB7.7 is highly specific for TB [34].

The present study has several limitations. Most significantly, it was conducted at a single referral center and included a relatively small number of cases. In particular, it is possible that the number of our study participants was insufficient to detect the difference between 2 tests, considering the test variability of the QFT-GIT [32,35] and QFT-Plus assays [36]. In addition, we compared only the differences in the IFN-γ values of the QFT-GIT and QFT-Plus assay before and after LTBI treatment, without using a placebo or observation arm. Moreover, the follow-up QFT tests were performed almost immediately after the end of LTBI treatment. It remained unclear whether we would have different results if the follow-up tests were checked at a later period.

In conclusion, we found no difference with respect to the change in IFN-γ levels before and after preventive treatment measured using the QFT-Plus assay and QFT-GIT assay in 44 participants with LTBI. Considering that the QFT-GIT assay is not a useful biomarker of response to LTBI treatment, the QFT-Plus also does not appear to be useful as a biomarker of LTBI treatment response.

## Supporting information

**S1 Fig. Changes in IFN-γ levels before and after preventive therapy.** IFN-γ concentrations from each participant are shown (•). Median levels are indicated by heavy lines. IFN-γ, interferon-γ; QFT-GIT, QuantiFERON-TB Gold In-Tube; QFT-Plus, QuantiFERON-TB Gold Plus.
(DOCX)

**S2 Fig. Comparison of IFN-γ response to QFT-Plus antigen TB1 and TB2 at baseline and after LTBI treatment.** (A) IFN-γ response to QFT-Plus antigen TB1 and TB2 is similar both at baseline and after LTBI treatment. (B) Positive correlation between IFN-γ values in response to TB1 and TB2 stimulation both at baseline and after LTBI treatment (Baseline: r = 0.912, $P < 0.001$, after LTBI treatment: r = 0.892, $P < 0.001$). IFN-γ, interferon-γ; QFT-Plus, QuantiFERON-TB Gold Plus; LTBI, latent tuberculosis infection.
(DOCX)

**S3 Fig. Changes in IFN-γ levels before and after preventive therapy according to the possibility of recent infection.** (A) The LTBI participants who were likely to have been infected recently; (B) the other participants. Difference in IFN-γ concentrations before and after preventive therapy for each participant are shown (•). Median levels are indicated by gray lines. IFN-γ, interferon-γ; QFT-GIT, QuantiFERON-TB Gold In-Tube; QFT-Plus, QuantiFERON-TB Gold Plus; LTBI, latent tuberculosis infection.
(DOCX)

**S4 Fig. Distribution of IFN-γ values before and after preventive therapy according to the baseline IFN-γ results.** IFN-γ concentrations from each participant are shown (•). IFN-γ, interferon-γ; QFT-Plus, QuantiFERON-TB Gold Plus.
(DOCX)

**S1 Table. Median IFN-γ levels before and after preventive therapy.**
(DOCX)

## Acknowledgments

We thank Qiagen for providing the QuantiFERON-TB Plus kits free of charge.

## Author Contributions

**Conceptualization:** Shinhee Park, Yong-Ha Jo, Mi-Na Kim, Heungsup Sung, Tae Sun Shim.

**Data curation:** Ock-Hwa Kim, Kyung-Wook Jo, Shinhee Park, Yong-Ha Jo, Mi-Na Kim, Heungsup Sung.

**Formal analysis:** Ock-Hwa Kim, Kyung-Wook Jo, Shinhee Park, Mi-Na Kim, Heungsup Sung.

**Investigation:** Ock-Hwa Kim, Kyung-Wook Jo, Tae Sun Shim.

**Methodology:** Yong-Ha Jo.

**Supervision:** Tae Sun Shim.

**Writing – original draft:** Ock-Hwa Kim, Kyung-Wook Jo.

**Writing – review & editing:** Kyung-Wook Jo, Tae Sun Shim.

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
