## [Decision Letter · Decision Letter 0]

17 Sep 2019

PONE-D-19-16329

Comparison of the change in QuantiFERON-TB Gold Plus and QuantiFERON-TB Gold In-Tube results after preventive therapy for latent tuberculosis infection

PLOS ONE

Dear Prof. Shim,

Thank you for submitting your manuscript to PLOS ONE. After careful consideration, we feel that it has merit but does not fully meet PLOS ONE’s publication criteria as it currently stands. Therefore, we invite you to submit a revised version of the manuscript that addresses the points raised during the review process.

Please, improve the paper according to the referee's comments (see below).

We would appreciate receiving your revised manuscript by Nov 01 2019 11:59PM. To enhance the reproducibility of your results, we recommend that if applicable you deposit your laboratory protocols in protocols.io, where a protocol can be assigned its own identifier (DOI) such that it can be cited independently in the future. For instructions see: http://journals.plos.org/plosone/s/submission-guidelines#loc-laboratory-protocols

We look forward to receiving your revised manuscript.

Kind regards,

Joan A Caylà, PhD, MD

Academic Editor

PLOS ONE

Journal Requirements:

Reviewers' comments:

Reviewer's Responses to Questions

**Comments to the Author**

1. Is the manuscript technically sound, and do the data support the conclusions?

Reviewer #1: Yes

Reviewer #2: Yes

2. Has the statistical analysis been performed appropriately and rigorously? 

Reviewer #1: I Don't Know

Reviewer #2: Yes

3. Have the authors made all data underlying the findings in their manuscript fully available?

Reviewer #1: No

Reviewer #2: Yes

4. Is the manuscript presented in an intelligible fashion and written in standard English?

Reviewer #1: Yes

Reviewer #2: Yes

5. Review Comments to the Author

Reviewer #1: The authors seek to examine whether the QFT plus before and after LTBI treatment was significantly different and if this could be used to as a biomarker for LTBI treatment response. A cohort of LTBI patients who had both the QFT-IT and the QFT-plus before and after LTBI treatment were examined in this analysis. Treatment of LTBI was managed as either 3 months of INH and rifampin or 4 months of rifampin. This area is novel as there have not been publications to address the use of QFT-Plus although the data for QFT-IT have not been promising. The data suggest that there is no overt evidence that the QFT-Plus can be used as a marker for LTBI treatment success. However, the manuscript lacks quite a bit of clinically relevant information that can be gleaned from the data set for the readership to appreciate. There are also important methodologic details that are missing that are outlined below.

Major issues

1. Was active TB excluded from those recruited into the LTBI study and how?

2. How long after treatment was the post treatment blood QFT draw performed?

3. It is worthwhile to have more descriptive components of the data. Was there ever discordance between the QFT-IT and the QFT-plus?

4. The statistical analysis in figure 2 (line 153) states that the p-value= 0.06 which is a trend. Was this the pre-hoc testing? Where does the statement, “ median IFN-g level value of the QFT-GIT assay was slightly higher than those of the QFT-PLUS assay” come from?

5. Figure 3 is unnecessary. Showing the log transformed data does not help clinicians as these numbers bear no significance and the data is already shown in Figure 2.

6. One of the main questions presented in the introduction was “to assess whether QFT-plus can be employed as a useful biomarker of LTBI treatment response”. While data in figure 4 shows the mean difference in IFN level before and after LTBI treatment, it would also be helpful to know if the overall mean between those before and after treatment were significantly different.

7. It is notable in figure 4 that there are some individuals that increase in IFN-g after treatment in the QFT-plus groups but not the QFT-IT. In general, were the trends between pre and post treatment similar between assays (i.e., when a reduction in IFN-g was observed by QFT-IT, was it there also an observed reduction in the QFT-Plus?) Were there any notable clinical issues between those who increased in IFN after treatment (e.g., recent exposure, treatment group as INH has been noted to increase during the course of treatment, time from ending treatment)?

Minor issues:

1. The introduction discusses the difference in TB1 and TB2 within the QFT plus assay and it is presumed that the QFTplus has both TB1 and TB 2 stimulation. Line 88 sates “those whose QFT-PLIS assay (TB1 or TB2) results…”. I presume that this should be written as (TB1 and TB2) but perhaps it is one or the other. Is one set of stimulators (TB 1 vs TB2) more likely to be positive then than the other based on the dataset given? And is one more likely to decrease than the other after treatment with LTBI?

2. In each figure, do the lines represent means or medians? If the data are not normally distributed, then it should be medians.

Reviewer #2: This is a straight-forward comparison of the response of 2 different versions of the Quantiferon test to treatment for latent tuberculosis infection. The response of the older version, QFT gold IT, has been previously tested and shown not to change significantly with LTBI treatment. The response of the newer version, QFT Plus, has been examined I a limited number of subjects. There are reasons to think that the QFT Plus may reflect an earlier CD 8 cell response and thus might be more influenced by LTBI treatment, thus, providing the rationale for the study. However the investigators found that the response did not change after treatment.

I think the study is sound, although the numbers are small, and the information is useful. However, my main observation is that the in the population studied there were only 5 subjects who might have been infected recently. The authors note that in an earlier study showed a significant decline in IFNG response following LTBI treatment and speculate that reason for the different results may be related to the fact that most of the subjects in the earlier study were likely to have been infected recently. Stronger evidence to this point is found in a study of tuberculin skin test responses following treatment with isoniazid in a group of US Navy personnel. The likelihood of skin test reversion to negative was strongly associated with the recency of infection.

(V. N. Houk MC, USN , D. C. Kent MC, USN , K. Sorensen MC, USNR

& J. H. Baker MC, USN (1968) The Eradication of Tuberculosis Infection by Isoniazid

Chemoprophylaxis, Archives of Environmental Health: An International Journal, 16:1, 46-50, DOI:

10.1080/00039896.1968.10665013)

I would suggest that the authors incorporate the findings from this study into their discussion in support of their speculation that recency of infection .influenced their findings

6. PLOS authors have the option to publish the peer review history of their article (what does this mean?). If published, this will include your full peer review and any attached files.

Reviewer #1: No

Reviewer #2: No

---

## [Author Response · Author response to Decision Letter 0]

5 Nov 2019

To the Editor of PLOS ONE

Dear Joan A Caylà:

We would like to thank you and the reviewers for the helpful comments pertaining to our manuscript entitled “Comparison of the change in QuantiFERON-TB Gold Plus and QuantiFERON-TB Gold In-Tube results after preventive therapy for latent tuberculosis infection” (manuscript ID: PONE-D-19-16329). We believe that the manuscript has been significantly improved after incorporation of the comments provided.

The manuscript has been revised in accordance with your suggestions. Our point-by-point responses to the reviewers’ comments are provided below.

Please note that we have changed the order of the authors (Ock-Hwa Kim and Kyung-Wook Jo) in the revised manuscript.

We hope that the manuscript is now suitable for publication in PLOS ONE.

Sincerely,

Tae Sun Shim, MD, PhD

Address: Division of Pulmonary and Critical Care Medicine,

University of Ulsan College of Medicine, Asan Medical Center,

88 Olympic-ro 43-gil, Songpa-gu, Seoul 05505, South Korea

Phone: 82-2-3010-3892; Fax: 82-2-3010-6968

E-mail: shimts@amc.seoul.kr

Responses to Reviewer 1:

The authors seek to examine whether the QFT plus before and after LTBI treatment was significantly different and if this could be used to as a biomarker for LTBI treatment response. A cohort of LTBI patients who had both the QFT-IT and the QFT-plus before and after LTBI treatment were examined in this analysis. Treatment of LTBI was managed as either 3 months of INH and rifampin or 4 months of rifampin. This area is novel as there have not been publications to address the use of QFT-Plus although the data for QFT-IT have not been promising. The data suggest that there is no overt evidence that the QFT-Plus can be used as a marker for LTBI treatment success. However, the manuscript lacks quite a bit of clinically relevant information that can be gleaned from the data set for the readership to appreciate. There are also important methodologic details that are missing that are outlined below.

Major issues

1. Was active TB excluded from those recruited into the LTBI study and how?

Thank you for your comments. We have added the following sentences in the Materials and methods section of the revised manuscript.

“All enrolled subjects were examined using the following strategy to rule out the possibility of active tuberculosis. All subjects were interviewed by the attending physician regarding the presence of any respiratory symptoms that were suggestive of active tuberculosis, such as cough, sputum, or hemoptysis. In addition, all patients underwent simple chest radiography. For those with radiographic anomalies suggestive of tuberculosis, further evaluation was performed using sputum acid-fast bacilli smear and culture to rule out active tuberculosis or lung disease due to nontuberculous mycobacteria. Moreover, present and former CXR findings were compared to identify the presence of newly developed lesions.”

(Page 4, lines 86–93 in the Methods section of the revised manuscript)

2. How long after treatment was the post treatment blood QFT draw performed?

Thank you for your comment. As mentioned in lines 182–183 of the original manuscript, follow-up QFT tests were performed at a median interval of 2 days (IQR 0–8.8) after the completion of LTBI treatment.

3. It is worthwhile to have more descriptive components of the data. Was there ever discordance between the QFT-IT and the QFT-plus?

Thank you for your insightful comment. As per your comment, we have analyzed if there was a discordance between the QFT-GIT and the QFT-Plus assays after LTBI treatment.

1) We have added following sentences in the Materials and methods section of the revised manuscript.

“To qualitatively compare the results of the QFT-GIT and QFT-Plus assays after LTBI treatment, positive, negative, and overall percent agreements were calculated using the results of QFT-GIT as the reference standard. Concordance between the QFT-GIT and QFT-Plus assays was assessed using kappa coefficients; kappa values of ≤0, 0.01–0.20, 0.21–0.40, 0.41–0.60, 0.61–0.80, and ≥0.81 were interpreted to indicate no, slight, fair, moderate, substantial, and almost perfect agreement, respectively [19].”

(Page 6, lines 134–139 in the Materials and methods section of the revised manuscript)

2) Furthermore, we have added the following sentences and Table 2 in the Results section of the revised manuscript.

“We qualitatively compared the results of the QFT-GIT and QFT-Plus assays after LTBI treatment. The QFT-Plus assay showed agreement with the QFT-GIT assay in 40/42 (95.2%) positive tests and 1/2 (50.0%) negative tests and had an overall agreement of 41/44 (93.2%) among all subjects, with a Cohen’s kappa value of 0.37 (fair agreement). Among the 44 subjects enrolled in our study, the QFT-GIT and QFT-Plus assays after LTBI treatment were discordant in 3 (6.8%) subjects (Table 2).” (Table 2 was in the attached file, "Response to reviewers".)

(Page 9, lines 195–page 10, line 203 in the Results section of the revised manuscript)

4. The statistical analysis in figure 2 (line 153) states that the p-value= 0.06 which is a trend. Was this the pre-hoc testing? Where does the statement, “median IFN-g level value of the QFT-GIT assay was slightly higher than those of the QFT-PLUS assay” come from?

Thank you for your valuable comment. 

1) Statistical analysis demonstrated in Figure 2 was a pre-hoc analysis using the Friedman test. We performed the Friedman test again and found that the P value was actually 0.040 but was miswritten as 0.060. We apologize for this error. There was a statistically significant difference in the IFN-γ levels measured in different antigen-containing tubes before LTBI treatment. The P value has been revised from 0.060 to 0.040 in the Abstract and Results sections of the revised manuscript as follows:

“The baseline IFN-γ levels were different among the three values; the median IFN-γ level with QFT-GIT, QFT-Plus TB1, and QFT-Plus TB2 was 3.395 IU/mL (interquartile range [IQR] 1.615–7.180), 3.060 IU/mL (IQR 0.655–7.045), and 2.880 IU/mL (IQR 0.668–7.170) (P = 0.040), respectively.”

(Page 2, lines 30–33 in the Abstract of the revised manuscript)

“Overall, statistical significance was noted among the three baseline IFN-γ values; that is, the median IFN-γ level of QFT-GIT TB antigen-nil levels was 3.395 IU/mL (IQR 1.615–7.180), the median TB1 QFT-Plus antigen IFN-γ level (TB1-Nil) was 3.060 IU/mL (IQR 0.655–7.045), and the median TB2 QFT-Plus antigen IFN-γ level (TB2-Nil) was 2.880 IU/mL (IQR 0.668–7.170) (P = 0.040).”

(Page 8, lines 162–166 in the Results section of the revised manuscript)

2) Post hoc analysis using Wilcoxon signed rank tests was conducted with the application of Bonferroni correction, resulting in a significance level of P < 0.017. There were no significant differences in the baseline IFN-γ values between QFT-GIT and QFT-Plus TB1 (Z = -1.743, P = 0.081) or between the QFT-GIT and QFT-Plus TB2 assays (Z = -1.367, P = 0.172).

We have added the following sentence concerning post hoc analysis to the Materials and methods section of the revised manuscript.

“In cases where there was a significant difference among the three values, post hoc analysis using Wilcoxon signed rank test for multiple comparisons was performed with the application of Bonferroni correction.”

(Page 6, lines 130–133 in the Materials and methods section of the revised manuscript)

We have revised following sentence by adding the results of the post hoc analysis in the Results section of the revised manuscript.

“The median IFN-γ level value of the QFT-GIT assay was slightly higher than that of the QFT-Plus assay; however, the difference was not statistically significant between the QFT-GIT and QFT-Plus TB1 (Z = -1.743, P = 0.081) or between QFT-GIT and QFT-Plus TB2 (Z = -1.367, P = 0.172) assays.”

(Page 8, line 166–169 in the Results of the revised manuscript)

5. Figure 3 is unnecessary. Showing the log transformed data does not help clinicians as these numbers bear no significance and the data is already shown in Figure 2.

Thank you for your comments. We completely agree with your opinion and we have deleted Figure 3 and the related sentences in the revised manuscript.

6. One of the main questions presented in the introduction was “to assess whether QFT-plus can be employed as a useful biomarker of LTBI treatment response”. While data in figure 4 shows the mean difference in IFN level before and after LTBI treatment, it would also be helpful to know if the overall mean between those before and after treatment were significantly different.

Thank you for your insightful comment. In addition to the median difference in the IFN-γ levels before and after LTBI treatment, we also re-analyzed whether the overall median IFN-γ levels before and after treatment were significantly different. We have represented the IFN-γ levels as median values because the data are not normally distributed.

 The median IFN-γ levels before and after preventive therapy

 QFT-GIT QFT-Plus TB1 QFT-Plus TB2

Before treatment 3.395 3.060 2.880

After treatment 6.804 2.905 3.880

As shown in the Table and Figure above (Table and Figure was in the attached file, "Response to reviewers".), the median IFN-γ levels measured using the QFT-GIT and QFT-Plus TB2 assays increased between baseline and the end of LTBI treatment (Wilcoxon signed rank test P = 0.008, 0.167, respectively). Although they were not statistically significant, the median IFN-γ levels of the QFT-Plus TB1 assay decreased after LTBI treatment (Wilcoxon signed rank test P = 0.247).

7. It is notable in figure 4 that there are some individuals that increase in IFN-g after treatment in the QFT-plus groups but not the QFT-IT. In general, were the trends between pre and post treatment similar between assays (i.e., when a reduction in IFN-g was observed by QFT-IT, was it there also an observed reduction in the QFT-Plus?) Were there any notable clinical issues between those who increased in IFN after treatment (e.g., recent exposure, treatment group as INH has been noted to increase during the course of treatment, time from ending treatment)?

Thank you for your comment.

1) Among the 44 subjects enrolled in our study, both the QFT-GIT and QFT-Plus (TB1 or TB2) assays showed similar trends (increase or decrease in IFN-γ) between pre- and post-treatment in 21 subjects. When an increase in IFN-γ levels was observed in 14 subjects by QFT-GIT, an increase in IFN-γ levels was also observed in the QFT-Plus assay.

2) The 14 subjects who showed an increase in IFN-γ levels after LTBI treatment in both QFT-GIT and QFT-Plus (TB1 or TB2) contained one subject with recently converted; however, none of the subjects in this group were in close contact with people with tuberculosis. All the subjects in this group were treated with isoniazid and rifampin for 3 months. The median duration from the completion of LTBI treatment to the follow-up QFT tests was 5.5 days (IQR 0.8–18.5). We could not find any notable clinical issues in the subjects with increased IFN-γ levels after LTBI treatment.

Minor issues:

1. The introduction discusses the difference in TB1 and TB2 within the QFT plus assay and it is presumed that the QFTplus has both TB1 and TB 2 stimulation. Line 88 sates “those whose QFT-PLIS assay (TB1 or TB2) results…”. I presume that this should be written as (TB1 and TB2) but perhaps it is one or the other. Is one set of stimulators (TB 1 vs TB2) more likely to be positive then than the other based on the dataset given? And is one more likely to decrease than the other after treatment with LTBI?



1) In the present study, the results of the QFT-Plus assay were defined as positive if either one (TB1 or TB2) or both of the TB antigen tubes were positive as outlined previously (J Clin Microbiol. 2018; 56(7): e00614-18). We had already described this in the original manuscript.

(Page 5, lines 104–105 in the Methods of the original manuscript)

2) 

Comparison of QFT-Plus TB1 and QFT-Plus TB2

 TB1 result

TB2 result Positive Negative

Positive 41 2

Negative 1 0

As shown in the Table above (Table was in the attached file, "Response to reviewers".), among the 44 subjects enrolled in our study, almost all patients (41/44, 93.2%) were positive with both QFT-Plus TB1 and TB2, with the exception of three subjects, one of whom was positive with QFT-Plus TB1 only, whereas two of whom were positive with QFT-Plus TB2 only.

The median differences in the IFN-γ levels using different antigen-containing tubes before and after LTBI treatment are shown in Figure 3 of the revised manuscript (Figure 4 in the original manuscript). There was no significant difference noted among the three values.

According to your comment #6, we also analyzed whether the overall median IFN-γ levels were significantly different before and after treatment. As shown in our response to comment #6, the median IFN-γ levels measured using the QFT-Plus TB2 assay tended to increase between baseline and the end of LTBI treatment (Wilcoxon signed rank test P = 0.167). However, the median IFN-γ levels of the QFT-Plus TB1 assay tended to decrease after LTBI treatment (Wilcoxon signed rank test P = 0.247).

2. In each figure, do the lines represent means or medians? If the data are not normally distributed, then it should be medians.

Thank you for your comment. The gray lines in each figure represent median values. We have added this information to each of the figure legends.

Responses to Reviewer 2:

This is a straight-forward comparison of the response of 2 different versions of the Quantiferon test to treatment for latent tuberculosis infection. The response of the older version, QFT gold IT, has been previously tested and shown not to change significantly with LTBI treatment. The response of the newer version, QFT Plus, has been examined I a limited number of subjects. There are reasons to think that the QFT Plus may reflect an earlier CD 8 cell response and thus might be more influenced by LTBI treatment, thus, providing the rationale for the study. However the investigators found that the response did not change after treatment.

I think the study is sound, although the numbers are small, and the information is useful. However, my main observation is that the in the population studied there were only 5 subjects who might have been infected recently. The authors note that in an earlier study showed a significant decline in IFNG response following LTBI treatment and speculate that reason for the different results may be related to the fact that most of the subjects in the earlier study were likely to have been infected recently. Stronger evidence to this point is found in a study of tuberculin skin test responses following treatment with isoniazid in a group of US Navy personnel. The likelihood of skin test reversion to negative was strongly associated with the recency of infection.

(V. N. Houk MC, USN , D. C. Kent MC, USN , K. Sorensen MC, USNR

& J. H. Baker MC, USN (1968) The Eradication of Tuberculosis Infection by Isoniazid

Chemoprophylaxis, Archives of Environmental Health: An International Journal, 16:1, 46-50, DOI:

10.1080/00039896.1968.10665013)

I would suggest that the authors incorporate the findings from this study into their discussion in support of their speculation that recency of infection influenced their findings

Thank you for your valuable comments. As you have correctly pointed out, the majority of the enrolled subjects in the present study were not patients who were likely to have been infected recently. Therefore, different results may have been observed if we studied recently infected patients, such as those in close contact with people with tuberculosis.

As the reviewer has noted, we have already described our opinion concerning these points in the Discussion section of the original manuscript as follows:

“Our findings also contrast with those of a recent study in Japan in which LTBI therapy significantly decreased IFN-γ levels and the number of responders to TB1- and TB2- peptide stimulation [24]. The most plausible explanation of this difference may be the difference of patient population. In that study, 80.3% (37/46) of patients with LTBI were recent contacts of patients with pulmonary tuberculosis, whereas the majority of our subjects were healthcare and day-care center workers or candidates for TNF inhibitor therapy (Table 1). In a previous study, approximately half of all contacts of tuberculosis patients with initial positive ELISPOT assay showed negative result on repeat ELISPOT assay at 3 months, in the absence of any treatment [27]. Therefore, the decline of TB1 and TB2 response on follow-up tests in tuberculosis contacts with positive QFT assays in previous study may be attributable to the natural course of IFN-γ response in a substantial proportion of patients, rather than to the effects of LTBI treatment. Further studies are needed to better elucidate this issue in a different study population.”

In the revised manuscript, we have added following sentences to this paragraph according to the reviewer’s suggestion.

“Our findings are in contrast with those of a recent study in Japan in which LTBI therapy significantly decreased IFN-γ levels and the number of responders to TB1- and TB2- peptide stimulation [25]. The most plausible explanation for this difference may be differences in the patient population. In the previous study, 80.3% (37/46) of patients with LTBI were recent contacts of patients with pulmonary tuberculosis, whereas the majority of subjects in our study were healthcare and day-care center workers or candidates for TNF inhibitor therapy. That is, only 11.4% (5/44, four subjects in close contact with people with tuberculosis and one subject with recently converted, Table 1) were likely to have been infected recently. Given that another previous study has also shown that the likelihood of tuberculin skin test reversion to negative following treatment with isoniazid in a group of US Navy personnel was strongly associated with the recency of infection [28], different results may have been observed if we enrolled patients who were likely to have been infected recently. In this context, the decline of the TB1 and TB2 response on follow-up tests in the subjects with positive QFT assays in the present study may be attributable to the natural course of IFN-γ response in a substantial proportion of patients, rather than to the effects of LTBI treatment. This speculation was supported by a previous study showing that approximately half of the contacts of tuberculosis patients with initial positive ELISPOT assay showed a negative result on repeat ELISPOT assay at 3 months, in the absence of any treatment [29]. Therefore, further studies are needed to better elucidate this issue in a different study population.”

(Page 11, lines 248–page 12, line 266 in the Discussion section of the revised manuscript)

---

## [Decision Letter · Decision Letter 1]

21 Apr 2020

PONE-D-19-16329R1

Comparison of the change in QuantiFERON-TB Gold Plus and QuantiFERON-TB Gold In-Tube results after preventive therapy for latent tuberculosis infection

PLOS ONE

Dear Prof. Shim,

Thank you for submitting your manuscript to PLOS ONE. After careful consideration, we feel that it has merit but does not fully meet PLOS ONE’s publication criteria as it currently stands. Therefore, we invite you to submit a revised version of the manuscript that addresses the points raised during the review process.

We would appreciate receiving your revised manuscript by Jun 05 2020 11:59PM. To enhance the reproducibility of your results, we recommend that if applicable you deposit your laboratory protocols in protocols.io, where a protocol can be assigned its own identifier (DOI) such that it can be cited independently in the future. For instructions see: http://journals.plos.org/plosone/s/submission-guidelines#loc-laboratory-protocols

We look forward to receiving your revised manuscript.

Kind regards,

Katalin Andrea Wilkinson, PhD

Academic Editor

PLOS ONE

Reviewers' comments:

Reviewer's Responses to Questions

**Comments to the Author**

1. If the authors have adequately addressed your comments raised in a previous round of review and you feel that this manuscript is now acceptable for publication, you may indicate that here to bypass the “Comments to the Author” section, enter your conflict of interest statement in the “Confidential to Editor” section, and submit your "Accept" recommendation.

Reviewer #1: All comments have been addressed

Reviewer #3: (No Response)

2. Is the manuscript technically sound, and do the data support the conclusions?

Reviewer #1: Yes

Reviewer #3: (No Response)

3. Has the statistical analysis been performed appropriately and rigorously? 

Reviewer #1: Yes

Reviewer #3: (No Response)

4. Have the authors made all data underlying the findings in their manuscript fully available?

Reviewer #1: Yes

Reviewer #3: (No Response)

5. Is the manuscript presented in an intelligible fashion and written in standard English?

Reviewer #1: Yes

Reviewer #3: (No Response)

6. Review Comments to the Author

Reviewer #1: There remain just a few minor revisions but otherwise the manuscript is acceptable for publication:

1. Statistics for figure 2- While the authors redid their statistics to find that there were differences in the results of baseline IFN-measurement between QFT-TB, TB1-QFT-plus and TB QFT plus. This would not be unusual given that the antigens used in each assay were different. While this fact may be fine for the results section of the manuscript, it may not be that critical for the abstract. Because the “objective was to assess whether QFT-Plus can serve as a biomarker of LTBI treatment response”, at a minimum the abstract should state either the number (and percentage) of positive results of QFT-PLUS and QFT-TB in the pre LTBI and post LTBI treatment groups to address the primary objective.

2. In the revised manuscript line 259-261, the authors state “the decline of the TB1 and TB2 responses on follow tests in the subjects with positive QFT assays in the present study” is not true. These were not statically significant and therefore this statement should be modified or removed from the revised manuscript.

Reviewer #3: Dear Katalin Andrea Wilkinson,

Thank you very much for sending this manuscript to me. This paper is worth publishing; however, some major revision is recommended.

Special attention to response to TB1 or TB2 peptides should consider.

Recent review articles in this issue should consider.

Can Interferon-γ Release Assays Be Useful for Monitoring the Response to Anti-tuberculosis Treatment?: A Systematic Review and Meta-analysis

Archivum Immunologiae et Therapiae Experimentalis volume 68, Article number: 4 (2020)

Comparison of the QuantiFERON-TB Gold Plus and QuantiFERON-TB Gold In-Tube interferon-γ release assays: A systematic review and meta-analysis

Advances in Medical Sciences. Volume 64, Issue 2, September 2019, Pages 437-443

How many had positive response to either TB1 or TB2 peptides?

The authors should stratify the QFT Plus results according to the ability of the subjects to respond to both TB1 and TB2 peptides (“TB1 and TB2”), only to TB1 (“only TB1”) or only to TB2 (“only TB2”).

The proportion of “only TB1” or “only TB2” responders change overtime or not?

Comparing the IFN-γ response to TB1 and TB2 peptides at the same time point is very important.

How was the median of TB1 and TB2 response at baseline and at the end of preventive therapy?

Did the IFN-γ production we stratified in LTBI subjects according to the time of exposure to Mtb?

In Petruccioli study, IFN-γ production decreased significantly in recent LTBI at the end of TB preventive therapy, in response to TB1 peptides. (see Effect of therapy on Quantiferon-Plus response in patients with active and latent tuberculosis infection Scientific reports | (2018) 8:15626)

it has been suggested the use of an uncertain zone for a better interpretation of the test resultsTo evaluate the distribution of IFN-γ values in the uncertain zone, authors should report their results stratifying the base-line results as follow: <0.2 IU/mL; ranging in 0.2–0.34 IU/mL; ranging in 0.35–0–7 IU/mL; >0.7 IU/Ml.

Effect of therapy on Quantiferon-Plus response in patients with active and latent tuberculosis infection Scientific reports | (2018) 8:15626

7. PLOS authors have the option to publish the peer review history of their article (what does this mean?). If published, this will include your full peer review and any attached files.

Reviewer #1: No

Reviewer #3: Yes: Shima Mahmoudi

---

## [Author Response · Author response to Decision Letter 1]

28 May 2020

To the Editor of PLOS ONE

Dear Katalin Andrea Wilkinson:

We would like to thank you and the reviewers for the helpful comments pertaining to our manuscript entitled “Comparison of the change in QuantiFERON-TB Gold Plus and QuantiFERON-TB Gold In-Tube results after preventive therapy for latent tuberculosis infection” (manuscript ID: PONE-D-19-16329R1). We believe that the manuscript has been significantly improved after incorporation of the comments provided.

The manuscript has been revised in accordance with your suggestions. Our point-by-point responses to the reviewers’ comments are provided below and attached as a file named "Response to reviewers".

We hope that the manuscript is now suitable for publication in PLOS ONE.

Sincerely,

Tae Sun Shim, MD, PhD

Address: Division of Pulmonary and Critical Care Medicine,

University of Ulsan College of Medicine, Asan Medical Center,

88 Olympic-ro 43-gil, Songpa-gu, Seoul 05505, South Korea

Phone: 82-2-3010-3892; Fax: 82-2-3010-6968

E-mail: shimts@amc.seoul.kr

Responses to Reviewer 1:

There remain just a few minor revisions but otherwise the manuscript is acceptable for publication:

1. Statistics for figure 2- While the authors redid their statistics to find that there were differences in the results of baseline IFN-measurement between QFT-TB, TB1-QFT-plus and TB QFT plus. This would not be unusual given that the antigens used in each assay were different. While this fact may be fine for the results section of the manuscript, it may not be that critical for the abstract. Because the “objective was to assess whether QFT-Plus can serve as a biomarker of LTBI treatment response”, at a minimum the abstract should state either the number (and percentage) of positive results of QFT-PLUS and QFT-TB in the pre LTBI and post LTBI treatment groups to address the primary objective.

Thank you for your valuable comments. We have deleted the sentence about the baseline IFN-γ levels and have added the following text, as well as Table 2, concerning the number (and percentage) of positive results of the QFT-Plus and QFT-GIT assays after LTBI treatment in the Abstract and the Results sections of the revised manuscript.

“Results: The mean age of the participants was 47.6 years. The QFT-GIT and QFT-Plus assays revealed positive results in 42/44 (95.5%) and 41/44 (93.2%) participants after LTBI treatment, showing overall agreement of 93.2%, with a Cohen’s kappa value of 0.37 (fair agreement). The differences between pre- and post-LTBI treatment interferon-γ levels were measured using the QFT-GIT and QFT-Plus assays. No significant differences were noted among the 3 values: the median difference in interferon-γ value with QFT-GIT, QFT-Plus TB1, and QFT-Plus TB2 was 0.211 IU/mL (IQR, −0.337–3.347), 0.025 IU/mL (IQR, −0.338–1.368), and 0.180 IU/mL (IQR, −0.490–2.278), respectively (P = 0.401).”

(Page 2, lines 33–35 in the Abstract of the revised manuscript)

“Table 2. The results of QFT assays at baseline and after the completion of preventive therapy.”

(Page 8–9, lines 168–172 in the Results section of the revised manuscript)

“After LTBI treatment, the QFT-GIT and QFT-Plus assays revealed positive results in 42/44 (95.5%) and 41/44 (93.2%) participants. Among the 44 participants with positive QFT-Plus assay results at baseline, 41/44 (93.2%) responded to both TB1 and TB2 at baseline, and the proportion did not change after treatment completion. In the remaining 3 participants, the QFT-Plus assay reverted to negative after LTBI treatment. The detailed results of the QFT-Plus according to the participants who responded to TB1 and/or TB2 peptides are shown in Table 2.”

(Page 9–10, lines 191–196 in the Results section of the revised manuscript)

2. In the revised manuscript line 259-261, the authors state “the decline of the TB1 and TB2 responses on follow tests in the subjects with positive QFT assays in the present study” is not true. These were not statically significant and therefore this statement should be modified or removed from the revised manuscript.

We have modified the text in the Discussion section of the revised manuscript as follows:

“Although we generally did not observe a statistically significant trend in terms of changes in IFN-γ level before and after LTBI treatment, some participants had a reduction in the TB1 and/or TB2 response on follow-up tests. This decline could be attributed to the natural course of IFN-γ response rather than to the effects of LTBI treatment, considering the findings of a previous study showing that approximately half of the contacts of patients with TB with initial positive ELISPOT assay had shown a negative result on repeat ELISPOT assay at 3 months, in the absence of treatment [29].”

(Page 14, lines 292–298 in the Discussion section of the revised manuscript)

Responses to Reviewer 3:

This paper is worth publishing; however, some major revision is recommended.

Special attention to response to TB1 or TB2 peptides should consider.

Recent review articles in this issue should consider.

1. How many had positive response to either TB1 or TB2 peptides?

Thank you for your valuable comments.

1) First of all, before we revise our present study, we want to highlight some of the differences between the present study and the study, “Effect of therapy on Quantiferon-Plus response in patients with active and latent tuberculosis infection,” on which you commented (Petruccioli E, Chiacchio T, Vanini V, et al. Effect of therapy on Quantiferon-Plus response in patients with active and latent tuberculosis infection. Sci Rep. 2018; 8: 15626.).

In the Petruccioli study, they have analyzed whether the overall median IFN-γ levels before and after treatment measured using only QFT-Plus were significantly different, using the Wilcoxon signed rank test.

However, the present study measured the median value of the differences between pre- and post-treatment IFN-γ to evaluate the change in each individual. Moreover, we compared the difference in the IFN-γ value measured using the QFT-Plus assay (TB1 or TB2) with the QFT-GIT result using the Friedman test.

2) Among the 44 participants enrolled in our study, most (41/44, 93.2%) were positive with both QFT-Plus TB1 and TB2, with the exception of 3 participants, 1 of who was positive with QFT-Plus TB1 only, whereas the other 2 were positive with QFT-Plus TB2 only.

We have thus added the following text to the Results section of the revised manuscript:

“Among the 44 participants, most (41/44, 93.2%) were positive with both QFT-Plus TB1 and TB2, with the exception of 3 participants, 1 of who was positive with QFT-Plus TB1 only, whereas the other 2 were positive with QFT-Plus TB2 only (Table 2).”

(Page 7, lines 154–157 in the Results section of the revised manuscript)

2. The authors should stratify the QFT Plus results according to the ability of the subjects to respond to both TB1 and TB2 peptides (“TB1 and TB2”), only to TB1 (“only TB1”) or only to TB2 (“only TB2”). The proportion of “only TB1” or “only TB2” responders change overtime or not?

We have stratified the study participants according to the results of the QFT-Plus assay with 2 different TB antigen tubes (TB1 and TB2).

Among the 44 participants with positive QFT-Plus assay results, 41/44 (93.2%) responded to both TB1 and TB2 at baseline, and the proportion did not change at the end of therapy. The proportion of only-TB1 and only-TB2 responders changed to 0% over time. In other words, 3 participants had QFT-Plus reversions after LTBI treatment. The participant (1/41, 2.4%) who had responded to both TB1 and TB2 and the other 2 (2/2, 100%) participants who had responded to only TB2 at baseline showed QFT-Plus reversion after LTBI treatment. The participant who had responded to only TB1 at baseline responded to both TB1 and TB2 at the end of therapy, and reversion was not observed.

We have added Table 2 and the following text to the Results section of the revised manuscript.

“Table 2. The results of QFT assays at baseline and after the completion of preventive therapy."

(Page 8–9, lines 168–172 in the Results section of the revised manuscript)

“Among the 44 participants with positive QFT-Plus assay results at baseline, 41/44 (93.2%) responded to both TB1 and TB2 at baseline, and the proportion did not change after treatment completion. In the remaining 3 participants, the QFT-Plus assay reverted to negative after LTBI treatment. The detailed results of the QFT-Plus according to the participants who responded to TB1 and/or TB2 peptides are shown in Table 2.”

(Page 9–10, lines 192–196 in the Results section of the revised manuscript)

3. Comparing the IFN-γ response to TB1 and TB2 peptides at the same time point is very important.

We have analyzed IFN-γ response to TB1 and TB2 stimulation at baseline and after LTBI treatment. Comparing the IFN-γ response to TB1 and TB2 peptides at the same time point, we observed similar levels of IFN-γ. In addition, we found a positive correlation between the IFN-γ values in response to TB1 and TB2 stimulation both at baseline and after LTBI treatment (Baseline: r = 0.912, P < 0.001; after LTBI treatment: r = 0.892, P <0.001), as described in the Petruccioli study.

We have added the following text and the Figure above to the Results section of the revised manuscript:

“In addition, when we compared the IFN-γ response to TB1 and TB2 peptides at the same time point, we observed similar levels of IFN-γ and a positive correlation between IFN-γ values in response to TB1 and TB2 stimulation both at baseline and after LTBI treatment. (S2 Fig).”

(Pages 11, lines 225–228 in the Results section of the revised manuscript)

4. How was the median of TB1 and TB2 response a baseline and at the end of preventive therapy?

In addition to the median difference in the IFN-γ levels before and after LTBI treatment, we also re-analyzed whether the overall median IFN-γ levels before and after treatment were significantly different. We have represented the IFN-γ levels as median values because the data were not normally distributed.

Median IFN-γ levels measured using the QFT-GIT and QFT-Plus TB2 assays increased between baseline and the end of LTBI treatment (Wilcoxon signed rank test P = 0.008, 0.167, respectively). Although they were not statistically significant, the median IFN-γ levels of the QFT-Plus TB1 assay decreased after LTBI treatment (Wilcoxon signed rank test P = 0.247).

We have added the following sentence, as well as the Figure and Table above, to the Results section of the revised manuscript.

“We also evaluated the overall median IFN-γ levels before and after treatment, the results of which are shown in the supporting information (S1 Fig and S1 Table).”

(Page 11, lines 223–225 in the Results section of the revised manuscript)

5. Did the IFN-γ production we stratified in LTBI subjects according to the time of exposure to Mtb? In Petruccioli study, IFN-γ production decreased significantly in recent LTBI at the end of TB preventive therapy, in response to TB1 peptides. (see Effect of therapy on Quantiferon-Plus response in patients with active and latent tuberculosis infection Scientific reports | (2018) 8:15626)

In the Petruccioli study, the subjects with LTBI were divided into those with remote infection (at least 3 years) and those with recent infection (no more than 3 months). However, as described at line 281–284 in the Discussion section of the revised manuscript, the majority of the participants in our study were healthcare and day care center workers or candidates for TNF inhibitor therapy, and we could not evaluate the time of exposure to M. tuberculosis.

Therefore, we divided the study participants into the those who were likely to have been infected recently (4 participants in close contact with people with tuberculosis and 1 recently converted) and the other participants, then observed IFN-γ production.

1) Among the 5 participants who were likely to have been infected recently, 4/5 (80.0%) showed a positive response in QFT-GIT and QFT-Plus (TB1 or TB2) after the completion of therapy, whereas in the other group, 35/36 (97.2%) and 34/36 (94.4%) responded after the treatment in QFT-GIT, and QFT-Plus (TB1 or TB2), respectively.

2) Analyzing the quantitative results, the median IFN-γ levels measured using the QFT-GIT assay, QFT-Plus TB1, and QFT-Plus TB2 increased between baseline and the end of LTBI treatment in the LTBI participants with probable recent infection (Wilcoxon signed rank test P = 0.686, 0.893, 0.686, respectively).

In the other group, the median IFN-γ levels measured using the QFT-GIT assay, QFT-Plus TB1, and QFT-Plus TB2 also increased between baseline and the end of LTBI treatment (Wilcoxon signed rank test P = 0.004, 0.248, 0.209, respectively) (See the Figures and Tables above).

3) In the LTBI participants who were likely to have been infected recently, the median difference of the IFN-γ value of QFT-GIT was −0.187 IU/mL (IQR, −2.131–0.895), the median difference of TB1 QFT-Plus antigen IFN-γ level was −0.080 IU/mL (IQR, −0.795–1.950), and the median difference of TB2 QFT-Plus antigen IFN-γ level was 0.050 IU/mL (IQR, −0.425–1.320) (P = 0.819).

In the other group, the median difference of the IFN-γ value of QFT-GIT was 0.228 IU/mL (IQR, −0.296–3.650), the median difference of TB1 QFT-Plus antigen IFN-γ level was 0.050 IU/mL (IQR, −0.330–1.370), and the median difference of TB2 QFT-Plus antigen IFN-γ level was 0.180 IU/mL (IQR, −0.510–2.370) (P = 0.454). 

We have added the following sentence and the Figures above to the Discussion section of the revised manuscript:

“Although the findings were generally the same when we stratified our participants according to the possibility of recent infection (S3 Fig), it was likely that the number of recently infected participants was too small to reveal statistical significance.”

(Page 14, line 288–291 in the Discussion section of the revised manuscript)

6. It has been suggested the use of an uncertain zone for a better interpretation of the test results. To evaluate the distribution of IFN-γ values in the uncertain zone, authors should report their results stratifying the base-line results as follow: <0.2 IU/mL; ranging in 0.2–0.34 IU/mL; ranging in 0.35–0–7 IU/mL; >0.7 IU/Ml. (see Effect of therapy on Quantiferon-Plus response in patients with active and latent tuberculosis infection Scientific reports | (2018) 8:15626)

Considering the inherent variability of the IGRA test, we performed our analysis again after stratifying our participants according to the baseline IFN-γ value of QFT-Plus (either TB1 or TB2), as previously described in the Petruccioli study: 0.2–0.34 IU/mL; 0.35–0.7 IU/mL; and >0.7 IU/mL. We found that 75% (33/44) of the participants had a baseline IFN-γ value >0.7 IU/mL in the QFT-Plus TB1 and TB2 assay, respectively.

Among 3 participants who had QFT-Plus reversions after LTBI treatment, 2 were in the uncertainty zone, which had been defined as baseline IFN-γ value 0.2–0.7 IU/mL in a previous study (Nemes E, Rozot V, Geldenhuys H, et al. Optimization and interpretation of serial QuantiFERON testing to measure acquisition of Mycobacterium tuberculosis infection. Am J Respir Crit Care Med. 2017; 196: 638–648.). The majority of the LTBI participants with IFN-γ values at baseline >0.7 IU/mL also had a response >0.7 IU/mL after the completion of preventive therapy.

We have added the following text, as well as the Figure above, to the Discussion section of the revised manuscript:

“Considering this inherent IGRA test variability, we performed our analysis again after stratifying our participants according to the baseline IFN-γ value of QFT-Plus (either TB1 or TB2) as previously described [25]: 0.2–0.34 IU/mL; 0.35–0.7 IU/mL; and >0.7 IU/mL. The overall results are shown in the supporting information (S4 Fig). Reversions tended to be more in the uncertainty zone, which had been defined as baseline IFN-γ value 0.2–0.7 IU/mL in previous studies [25,33]. This result suggests that some of the reversions could be attributable to inherent variability.”

(Page 14–15, lines 308–314 in the Discussion section of the revised manuscript)

---

## [Editor Report · Decision Letter 2]

2 Jun 2020

Comparison of the change in QuantiFERON-TB Gold Plus and QuantiFERON-TB Gold In-Tube results after preventive therapy for latent tuberculosis infection

PONE-D-19-16329R2

Dear Dr. Shim,

We are pleased to inform you that your manuscript has been judged scientifically suitable for publication and will be formally accepted for publication once it complies with all outstanding technical requirements.

With kind regards,

Katalin Andrea Wilkinson, PhD

Academic Editor

PLOS ONE
---

## [Editor Report · Acceptance letter]

5 Jun 2020

PONE-D-19-16329R2 

Comparison of the change in QuantiFERON-TB Gold Plus and QuantiFERON-TB Gold In-Tube results after preventive therapy for latent tuberculosis infection 

Dear Dr. Shim:

I'm pleased to inform you that your manuscript has been deemed suitable for publication in PLOS ONE. Congratulations! Your manuscript is now with our production department. 

Kind regards, 

on behalf of

Associate Professor Katalin Andrea Wilkinson 

Academic Editor

PLOS ONE